# Validity of Items Assessing Self-Reported Number of Breaks in Sitting Time among Children and Adolescents

**DOI:** 10.3390/ijerph17186708

**Published:** 2020-09-15

**Authors:** Veerle Van Oeckel, Benedicte Deforche, Nicola D. Ridgers, Elling Bere, Maïté Verloigne

**Affiliations:** 1Department of Public Health and Primary Care, Faculty of Medicine and Health Sciences, Ghent University, 9000 Ghent, Belgium; benedicte.deforche@ugent.be (B.D.); maite.verloigne@ugent.be (M.V.); 2Institute for Physical Activity and Nutrition (IPAN), School of Exercise and Nutrition Sciences, Deakin University, Melbourne, VIC 3125, Australia; nicky.ridgers@deakin.edu.au; 3Department of Public Health, Sport and Nutrition, Faculty of Sport, Science and Physical Education, University of Agder, 4604 Kristiansand, Norway; elling.bere@uia.no

**Keywords:** child, adolescent, sedentary behaviour, activPAL, surveys and questionnaires, psychometrics

## Abstract

*Background*: Sedentary behaviour guidelines recommend that individuals should regularly break up sitting time. Accurately monitoring such breaks is needed to inform guidelines concerning how regularly to break up sitting time and to evaluate intervention effects. We investigated the concurrent validity of three “UP4FUN child questionnaire” items assessing the number of breaks in sitting time among children and adolescents. *Methods*: Fifty-seven children and adolescents self-reported number of breaks from sitting taken at school, while watching TV, and during other screen time activities. Participants also wore an activPAL monitor (PAL Technologies, Glasgow, UK) to objectively assess the number of sitting time breaks (frequency/hour) during the school period and the school-free period (which was divided in the periods “after school” and “during the evening”). Concurrent validity was assessed using Spearman rank correlations. *Results*: Self-reported number of breaks/hour at school showed good concurrent validity (ρ = 0.676). Results were moderate to good for self-reported number of breaks/hour while watching TV (ρ range for different periods: 0.482 to 0.536) and moderate for self-reported number of breaks/hour in total screen time (ρ range for different periods: 0.377 to 0.468). Poor concurrent validity was found for self-reported number of breaks/hour during other screen time activities (ρ range for different periods: 0.157 to 0.274). Conclusions: Only the questionnaire items about number of breaks at school and while watching TV appear to be acceptable for further use in research focussing on breaks in prolonged sitting among children and adolescents.

## 1. Introduction

Sedentary behaviour is defined as “any waking behaviour characterised by an energy expenditure ≤ 1.5 metabolic equivalents (MET’s) while in a sitting, reclining, or lying posture” [1]. School-aged children are sedentary for approximately eight hours per day [2]. The school context contributes significantly to this behaviour, as children and adolescents spend more than 65% of the time at school sedentary [3,4]. Findings on the consequences of higher volumes and more prolonged bouts of sedentary behaviour among children and adolescents are less consistent compared to evidence in adults [5,6,7]. Nonetheless, there is evidence in children and adolescents to suggest that higher levels of sedentary behaviour, especially screen-based behaviour [8,9], are associated with cardiometabolic risk factors [8,9,10]; behavioural problems, aggression, and inattention [8]; lower fitness [8,10]; lower self-esteem [8] and poorer health-related quality of life [11]. Furthermore, there are several studies in children and adolescents showing that prolonged sitting is associated with an increased cardiometabolic disease risk [12], a more unfavourable body mass index [13] and fat mass index score [13], and decreased physical fitness [14]. However, other reviews state that there is limited evidence available of an association between prolonged sedentary time and cardiometabolic health in children and adolescents [5,7]. Nevertheless, next to reducing total time spent sedentary, sedentary behaviour guidelines also recommend breaking up long periods of sitting as often as possible [15,16].

To inform specific guidelines concerning how regularly to break up sitting time and to evaluate intervention effects, the ability to assess breaks from sitting time is needed. Objective measurement tools, such as accelerometers and inclinometers, are considered to be valid and reliable instruments for measuring breaks in sedentary time in children and adolescents [17]. The low compliance rates are often mentioned as a limitation for these tools, particularly for the activPAL in youth due to participant burden and the negative side effects of monitor wear [18]. Furthermore, objective measurement tools have been less feasible for studies with large populations as this is a relatively expensive and time-consuming method [19], although a number of large population studies have collected device-based data (e.g., UK Biobank [20]). Therefore, there is a need for self-report tools that assess breaks in sitting time as they are relatively inexpensive and easy to administer [21,22]. However, the presence of recall bias and the possibility for participants to respond in a socially desirable manner are the main limitations of self-report tools [21]. To date, the reliability and validity of self-report questions related to breaks in sedentary time have only been investigated in adults. Two such studies have evaluated the reliability and validity of self-reported frequency of breaks from sitting at work and found poor [23] and adequate [24] test–retest reliability. In addition, concurrent validity estimates were poor [23] and fair [24] against inclinometer (activPAL) measures. There is clearly a need to examine the psychometric properties of questionnaires examining the frequency of sitting breaks in children and adolescents. One such questionnaire, the “UP4FUN child questionnaire”, was designed to assess the frequency of breaks from sitting time in children [25]. This questionnaire was used to evaluate the effect of the UP4FUN intervention as part of the ENERGY project (a European project to prevent overweight and obesity in 10 to 12-year-old children) with the UP4FUN intervention aiming to reduce total sedentary time and to break up sitting time regularly in 10 to 12-year-old children [26]. The test–retest reliability of the “UP4FUN child questionnaire” has already been assessed [26]. The items related to breaking up sedentary behaviour showed an intraclass correlation coefficient ranging from 0.68 to 0.72. However, the validity has yet to be investigated.

This study aimed to determine the concurrent validity of the items from the “UP4FUN child questionnaire” assessing the self-reported number of breaks in sitting time against objectively measured breaks in sitting time among children and adolescents. 

## 2. Materials and Methods 

This study used baseline data from a cluster-randomised controlled trial which evaluated the effect of standing desks in primary and secondary school classes in Flanders [27]. A convenience sample of 26 primary and secondary schools was invited to participate in the study. From this sample, 19 schools agreed to participate (ten primary schools, nine secondary schools). In each school, the school principal randomly selected one class (a 5th grade class with most pupils being aged 10–11 years old for primary schools and a 10th grade class with most pupils being aged 15–16 years old for secondary schools). A random subsample of three pupils per class (*n* = 57) completed a questionnaire and simultaneously wore an activPAL monitor to objectively assess number of breaks in sitting time. All subjects gave their informed consent for inclusion before they participated in the study. The study was conducted in accordance with the Declaration of Helsinki, and the study protocol was approved by the ethics committee of the Ghent University Hospital (B670201628738).

### 2.1. Measures

#### 2.1.1. Questionnaire

All children and adolescents completed a questionnaire in their classroom [27]. The “UP4FUN child questionnaire” items that were relevant for this study are the number of breaks in sitting time during a normal lesson at school, during one hour of watching TV, and during one hour of other screen time activities on a usual day. These items are displayed in Table 1. In addition to the three original items about the number of breaks in sitting time, a fourth variable was calculated: “number of breaks per hour in total screen time during leisure time”. This is the average value of “the number of breaks per hour while watching TV” and “the number of breaks per hour during other screen time activities” (i.e., both variables were summed and divided by two). As the questionnaire items report the number of breaks during one full hour of a particular activity, the average of the two items was taken to maintain the number of breaks per hour as the outcome.

#### 2.1.2. activPAL

Due to the limited availability of activPAL (PAL Technologies, Glasgow, UK) monitors combined with a large number of schools that had measurements in a short period, a subsample of three pupils per class wore an activPAL monitor for four or five school days using a 24 h protocol. Teachers were asked to randomly select three pupils. The devices were distributed on Monday or Tuesday and were collected on Friday afternoon. The device was attached to the anterior midline of the right thigh using 3M^TM^ Tegaderm Transparent Film Roll (3M, St. Paul, MN, USA). The activPAL monitor is a valid measure for estimating time spent sitting, standing and walking in children [17]. The activPAL data were initially downloaded in 15-s epoch files using activPAL3™ software (v7.2.32) and were processed using a customised Excel macro (Deakin University, Melbourne, Australia). Non-wear time was calculated as periods of more than 20 min of consecutive zero counts [28]. Pupils who had at least 2 weekdays of a minimum of nine hours wear time per day were included. Since sleep should not be considered as sedentary behaviour, overnight data were distinguished from daytime data by determining a fixed sleep period for primary (from 9 p.m. to 7 a.m.) and secondary (from 10:30 p.m. to 7 a.m.) school pupils. This is consistent with previous studies in youth [29]. Wear time (expressed in minutes) and the number of sit-to-stand transitions were derived from the activPAL data, using transitions from a sitting posture to an upright posture as the outcome for the frequency of breaks from sitting. These two variables were calculated for the period at school and for the school-free period. The same start and end time was used for all schools [27]. In the included schools, the school day started between 8:30 a.m. and 8:40 a.m. and ended between 3:35 p.m. and 4:20 p.m. Therefore, the period at school was defined between 8:35 a.m. and 4 p.m. The school-free period was divided in the “after school period” (4 p.m. to 6 p.m.) and “evening period” (6 p.m. to 9 p.m. for primary school children; 6 p.m. to 10:30 p.m. for secondary school children) [30]. The number of sit-to-stand transitions in a specific period was transformed to the number of sit-to-stand transitions per hour for that period, for comparison with the questionnaire items about breaking up sitting time, which also reported number of breaks per hour. This was calculated by dividing the number of sit-to-stand transitions in a period by the wear time of that period in hours.

### 2.2. Data Analysis

Data were analysed using IBM SPSS Statistics for Windows, version 25 (IBM Corp., Armonk, NY, USA). Since the questionnaire data were ordinally scaled, validity was assessed using non-parametric tests. Spearman rank correlation coefficients were calculated to assess the concurrent validity of self-reported number of breaks per hour in sitting time against the objective number of sit-to-stand transitions per hour (activPAL) of the corresponding period(s). Spearman’s Rho correlation coefficients were interpreted as: “low” (<0.3); “moderate” (0.30–0.50); or “high” (>0.50) [31]. The data file associated with this article is available online at Mendeley Data in Appendix A.

## 3. Results

Table 2 describes the characteristics of the participants with valid activPAL and questionnaire data at the baseline test (*n* = 48). 

Table 3 shows the concurrent validity results. Good agreement was found between self-reported number of breaks/hour at school and number of activPAL sit-to-stand transitions/hour at school (ρ = 0.676). Self-reported number of breaks/hour while watching TV showed good agreement with number of activPAL sit-to-stand transitions/hour in the school-free period (ρ = 0.536), whereas a moderate agreement was found with activPAL sit-to-stand transitions/hour after school (ρ = 0.482) and during the evening (ρ = 0.495). Poor agreement was found between self-reported number of breaks/hour during other screen time activities and number of activPAL sit-to-stand transitions/hour in the school-free period (ρ = 0.228), after school (ρ = 0.157), and during the evening (ρ = 0.274). Moderate agreement was found between self-reported number of breaks/hour in total screen time and number of activPAL sit-to-stand transitions/hour in the school-free period (ρ = 0.468), after school (ρ = 0.377), and during the evening (ρ = 0.465).

## 4. Discussion

This study aimed to investigate the concurrent validity of self-report items assessing number of breaks in sitting time among children and adolescents. Results indicated poor to good concurrent validity for these items. A better concurrent validity was observed for breaks in sitting time at school compared to outside of school. 

This study found that concurrent validity estimates were highest for self-reported number of breaks at school. Since the school period is relatively structured and breaks during school lessons might be more planned at fixed times, it might be easier to recall the number of breaks taken. In addition, as children accumulate more sitting time and break up sitting less frequently within school time when compared with non-school time [3], it is probably easier to recall breaks during school time. Further, concurrent validity estimates were consistently higher for self-reported number of breaks while watching TV compared to self-reported number of breaks during other screen time activities. It might be that breaks while watching TV are also somewhat easier to estimate since this behaviour may be more clearly defined, whereas other screen time activities (e.g., smartphone use) may be more fragmented. When comparing our concurrent validity estimates with past research, the study of Sudholz, Ridgers, Mussap, Bennie, Timperio, and Salmon [24] reported fair agreement between self-reported number of breaks in sitting per work hour and activPAL data in adults (ρ = 0.39). These results are quite comparable to our findings. However, another study found a poor correlation between activPAL measurements and the self-reported number of breaks in occupational sitting (ρ = 0.06) and while watching TV (ρ = 0.09) in adults [23]. These poor values were attributed to substantial missing data for these items. In addition, it is relevant to mention that the latter study assessed the self-reported number of breaks per day [23], whereas the study of Sudholz, Ridgers, Mussap, Bennie, Timperio, and Salmon [24] and our study assessed the self-reported number of breaks per hour. The results of our study and previous research indicate that the number of breaks in sitting time might be difficult to recall, so using a shorter time frame might be relevant, although this requires further investigation.

This study found acceptable concurrent validity for the items on the number of breaks in sitting time at school and while watching TV, despite limitations concerning the congruence between the self-reported number of breaks in sitting time during a specific behaviour (i.e., sitting during a lesson, while watching TV, and during other screen time activities) and the number of activPAL sit-to-stand transitions during the different periods (i.e., during an average hour of the entire school period and during an average hour of the entire school-free period). This suggests that other activities than sitting could have taken place during those periods. However, the congruence is somewhat better between the self-reported number of breaks in sitting time at school and the school period extracted from the activPAL data compared to the other questionnaire items and the activPAL periods they are correlated with, which might also explain the better concurrent validity for the self-reported number of breaks at school. The incongruence might partly explain the discrepancy in the number of breaks between the questionnaire items and the activPAL (e.g., 1.25 breaks/hour at school reported in the questionnaire compared to 4.81 sit-to-stand transitions/hour at school recorded by the activPAL). Another explanation for this discrepancy is that self-report measures can only assess conscious breaks in sitting time, compared to objective measurements that assess all breaks in sitting time (both conscious and unconscious ones). As a result, the specific items about the number of breaks in sitting time at school and while watching TV are only suited when aiming at measuring conscious breaks in sitting time (e.g., as intervention outcome) or when the purpose is to rank people according to their number of breaks in sitting time (e.g., determining whether someone breaks up sitting time more frequently compared to others). These questionnaire items might not be suited for research aiming to accurately assess the total number of breaks in sitting time.

Another important point is that the maximum number of breaks that could be recorded by the questionnaire was four breaks per hour, which is lower than the average number of breaks per hour recorded by the activPAL for the different periods (4.81 breaks/hour at school, 5.36 breaks/hour in the school-free period, 5.61 breaks/hour in the after school period and 5.23 breaks/hour during the evening). This might have biased our concurrent validity estimates, although the impact on our results will probably be limited as a relatively small proportion of the participants answered to break up sitting time four times or more per hour (for the number of breaks in sitting time at school: 8.3%; while watching TV: 20.8%; during other screen time activities: 20.8%). However, in future research, providing more response categories including a higher number of breaks should be explored. 

This study has some limitations. First, no a priori power analyses were conducted. As a result, our sample size was too small to examine whether the concurrent validity of the self-report measure differed according to age and sex. Further research in larger samples is needed to address this knowledge gap. Secondly, selection bias could have occurred when the school principal selected one class to participate in the study and when teachers selected three pupils to wear an activPAL monitor.

To the best of our knowledge, this is the first study that examines the validity of questionnaire items assessing self-reported number of breaks in sitting time in children and adolescents, which is a major strength of our study. Another strength relates to the use of the activPAL inclinometer as a criterion measure. The activPAL is an objective and valid measure for classifying posture as either sitting or standing in children and adolescents [17].

## 5. Conclusions

The items from the ‘UP4FUN child questionnaire’ about the number of breaks in sitting time at school and while watching TV appear to be acceptable for further use in research among children and adolescents. Higher concurrent validity estimates were observed for breaks in sitting time at school compared to the school-free period.

## Figures and Tables

**Table 1 ijerph-17-06708-t001:** Overview of included questionnaire items and their response categories.

Variable	Questionnaire Item	Answer Alternatives
Number of breaks in sitting time at school	During a normal school lesson, how often do you usually stand up, stretch or walk around a bit?	0 = Never, 1 = Once, 2 = Twice, 3 = 3 times, 4 = 4 times or more.
Number of breaks while watching TV	During one hour of watching TV/DVD, how often do you usually stand up, stretch or walk around a bit?	0 = Never, 1 = Once, 2 = Twice, 3 = 3 times, 4 = 4 times or more.
Number of breaks during other screen time activities	During one hour of using a computer/games console/smartphone for leisure activities, how often do you usually stand up, stretch or walk around a bit?	0 = Never, 1 = Once, 2 = Twice, 3 = 3 times, 4 = 4 times or more.

**Table 2 ijerph-17-06708-t002:** Characteristics of participants with valid data.

Demographics	
Sample size (*n*)	48
Mean age primary school pupils (Years;months ± SD)	10;7 ± 0;4
Mean age secondary school pupils (Years;months ± SD)	15;6 ± 0;3
Male (*n* (%))	23 (47.9%)
Primary school pupils (*n* (%))	25 (52.1%)
Self-reported number of breaks in sitting time (Mean ± SD; ranging from 0 to 4)	
Number of breaks/hour at school	1.25 ± 1.45
Number of breaks/hour while watching TV	1.95 ± 1.52
Number of breaks/hour during other screen time activities	2.08 ± 1.40
Number of breaks/hour in total screen time	2.00 ± 1.27
activPAL	
Number of valid days the activPAL was worn (Mean ± SD)	3.54 ± 0.87
Wear time in minutes per day (Mean ± SD)	808.73 ± 45.02
activPAL number of sit-to-stand transitions (Mean ± SD; continuous variable)	
Number of sit-to-stand transitions/hour at school	4.81 ± 1.81
Number of sit-to-stand transitions/hour in the school-free period	5.36 ± 2.44
Number of sit-to-stand transitions/hour after school	5.61 ± 2.67
Number of sit-to-stand transitions/hour during the evening	5.23 ± 2.52

**Table 3 ijerph-17-06708-t003:** Spearman’s Rho as indicator of concurrent validity of self-reported breaking up against activPAL sit-to-stand transitions.

Questionnaire Variable	activPAL	Spearman’s Rho [95% C.I.]
Number of breaks/hour at school	Number of sit-to-stand transitions/hour at school	0.676 [0.434–0.827]
Number of breaks/hour while watching TV	Number of sit-to-stand transitions/hour in the school-free period	0.536 [0.249–0.737]
	Number of sit-to-stand transitions/hour after school	0.482 [0.183–0.699]
	Number of sit-to-stand transitions/hour during the evening	0.495 [0.199–0.708]
Number of breaks/hour during other screen time activities	Number of sit-to-stand transitions/hour in the school-free period	0.228 [−0.113–0.521]
	Number of sit-to-stand transitions/hour after school	0.157 [–0.183–0.463]
	Number of sit-to-stand transitions/hour during the evening	0.274 [−0.066–0.577]
Number of breaks/hour in total screen time	Number of sit-to-stand transitions/hour in the school-free period	0.468 [0.167–0.689]
	Number of sit-to-stand transitions/hour after school	0.377 [0.063–0.623]
	Number of sit-to-stand transitions/hour during the evening	0.465 [0.163–0.687]

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
