# Peer review of "Validity of Items Assessing Self-Reported Number of Breaks in Sitting Time among Children and Adolescents"

_ijerph, 2020, doi:10.3390/ijerph17186708_

Round 1

Reviewer 1 Report

It does not show great attractiveness in terms of reading, although it may be useful in methodological terms for future investigations

Reviewer 2 Report

ijerph-841595

The study assessed the concurrent validity of a self-report instrument to assess frequency of breaks in sitting in children and adolescents. The paper is well written and provides some evidence that selected items from the instrument may have some usefulness. Paper is limited by the sample size and inability to examine how factors such as age and sex may influence the correlation between the two measures. Several comments below.

Detail the reason for using the average of the two items to represent number of breaks in total screen time rather than the sum of the two items. Potentially due to overlap in screen behaviours but please clarify.

Further detail is needed on the activpal protocol. Was 24 hr protocol used, if so how were sleep times dealt with. Consider make the excel spreadsheet and sample (de-identified) data file available as supplement for transparency. Did ‘school’ periods reflect actual school/class times?

Withstanding the limitations of the response scale in the survey items are additional measures that can provide insight on the agreement or mismatch between the two measures possible. Eg weighted kappa or other suitable measures given the data  

Clarify if items 2-3 had a defined recall period in terms of normal day similar to the school time item.

Line 184 -200 Given the observed correlation coefficients are broadly comparable with other studies, and only two of the self-report measures achieved ‘high’ ratings suggest toning down and clarifying (by stating which items rather than the overall instrument) has what may be considered acceptable concurrent validity.

on reading the abstract it is unclear why values are reported as ranges, please clarify what the ranges refer to (eg on multiple days?)

authors note that activpal assessed any breaks, whereas self-report may have assessed longer breaks in sitting. Did the authors consider using a minimum duration criteria for a break and how this influenced observations?

Also note in limitations that only the correlation between the measures was examined and not agreement.

should abstract specify the details (name , school or non school day) of the self-reprot instrument

intro L57-59. Note that this is changing with studies such as UK Biobank, NHANES and other large studies.

L73. Detail the test retest coefficients here to provide context as to what is good

L166. Break up sitting less frequently?

L188. Clarify this statement do you mean short period in terms of break per hour, or as item 1 uses the normal school day do you mean in the last week/ day.

Post hoc power calculations aren’t that useful and the statement about this in the limitations isn’t clear, including the basis of calculation. Suggest omitting, and state that a priori power was not determined and that study is limited by small sample.

Reviewer 3 Report

Good job, but the results should improve (years without decimals, sex in frequency and not%, ...)

Round 2

Reviewer 2 Report

The authors have addressed the comments adequately , no further comments.